# IDO1 Protein Is Expressed in Diagnostic Biopsies from Both Follicular and Transformed Follicular Patients

**DOI:** 10.3390/ijms24087314

**Published:** 2023-04-15

**Authors:** Marie Beck Hairing Enemark, Emma Frasez Sørensen, Trine Engelbrecht Hybel, Maja Dam Andersen, Charlotte Madsen, Kristina Lystlund Lauridsen, Bent Honoré, Francesco d’Amore, Trine Lindhardt Plesner, Stephen Jacques Hamilton-Dutoit, Maja Ludvigsen

**Affiliations:** 1Department of Hematology, Aarhus University Hospital, 8200 Aarhus, Denmark; mariem@rm.dk (M.B.H.E.); emmase@rm.dk (E.F.S.); trihyb@rm.dk (T.E.H.); maja.aner@auh.rm.dk (M.D.A.); frandamo@rm.dk (F.d.); 2Department of Clinical Medicine, Aarhus University, 8000 Aarhus, Denmark; 3Department of Pathology, Aarhus University Hospital, 8200 Aarhus, Denmark; krislaur@rm.dk (K.L.L.); stephami@rm.dk (S.J.H.-D.); 4Department of Biomedicine, Aarhus University, 8000 Aarhus, Denmark; bh@biomed.au.dk; 5Department of Pathology, Copenhagen University Hospital, 2100 Copenhagen, Denmark; trine.lindhardt.plesner@regionh.dk

**Keywords:** follicular lymphoma, histological transformation, indoleamine 2,3-dioxygenase

## Abstract

Follicular lymphoma (FL) is a lymphoid neoplasia characterized by an indolent clinical nature. Despite generally favorable prognoses, early progression and histological transformation (HT) to a more aggressive lymphoma histology remain the leading causes of death among FL patients. To provide a basis for possible novel treatment options, we set out to evaluate the expression levels of indoleamine 2,3-dioxygenase 1 (IDO1), an immunoinhibitory checkpoint molecule, in follicular and transformed follicular biopsies. The expression levels of IDO1 were assessed using immunohistochemical staining and digital image analysis in lymphoma biopsies from 33 FL patients without subsequent HT (non-transforming FL, nt-FL) and 20 patients with subsequent HT (subsequently transforming FL, st-FL) as well as in paired high-grade biopsies from the time of HT (transformed FL, tFL). Despite no statistical difference in IDO1 expression levels seen between the groups, all diagnostic and transformed lymphomas exhibited positive expression, indicating its possible role in novel treatment regimens. In addition, IDO1 expression revealed a positive correlation with another immune checkpoint inhibitor, namely programmed death 1 (PD-1). In summary, we report IDO1 expression in all cases of FL and tFL, which provides the grounds for future investigations of anti-IDO1 therapy as a possible treatment for FL patients.

## 1. Introduction

Follicular lymphoma (FL) is an indolent B-cell malignancy that comprises approximately 20% of adult non-Hodgkin lymphomas in Western countries [1]. The introduction of the CD20-targeting antibody, rituximab, in standard FL regimen has improved treatment efficacy and, as a consequence, the median survival for newly diagnosed FL patients is approaching several decades. However, the disease is generally incurable, and a substantial proportion of patients remain at risk of adverse outcomes with early progression, treatment refractoriness, and histological transformation to a more aggressive lymphoma histology [2,3,4]. The clinical course may vary greatly when comparing individual FL patients, with alternating periods of slow progression and varying degrees of generalized symptoms, in which various therapeutic interventions may be considered [4,5]. With clinical and biological heterogeneity increasingly being recognized, the incurable aspect makes FL a challenging malignancy to manage [6]. Despite the advances in FL treatment, early progression and histological transformation remain the leading causes of FL-related mortality [7,8,9]. Thus, there is a need for additional personalized therapies.

To provide a basis for possible treatment options that are more extensive, we evaluated the expression levels of indoleamine 2,3-dioxygenase 1 (IDO1) in diagnostic FL biopsies. IDO1 catalyzes the first and rate-limiting step of tryptophan (Trp) catabolism along the kynurenine (kyn) pathway [10]. The depletion of Trp into kyn leads to important immunosuppressive functions by activating regulatory T-cells (Tregs) and myeloid-derived suppressor cells, ultimately suppressing the function of immune effector cells [11,12]. Given its biological importance, IDO1 has become an attractive target in cancer therapy, with initiated clinical trials investigating different IDO inhibitors [12,13,14]. In various cancers, IDO1 is expressed both in tumor cells and in cancer-associated cells in the tumor microenvironment, including antigen-presenting cells [12]. Previous studies have shown both CD4+ and CD8+ T-cell reactivity towards IDO-derived peptides [11,15,16]. As a result, clinical studies employing IDO-inhibitors as well as an IDO1 peptide-based vaccine strategy for cancer treatment have been initiated [12,13,14]. Effector cell activation through IDO vaccination may not only eliminate the antigen-expressing tumor cells but may also induce immunological activation through the simultaneous enhancement of other effector responses and reduced regulatory signals [12,13,17]. The current treatment of FL and transformed FL generally includes chemotherapy and/or immunotherapy, often in combination with CD20-targeting rituximab. Thus, as anti-IDO therapies are already emerging, this might be a novel target in the future treatment of FL. Therefore, in the present study, we aimed to describe IDO1 expression in lymphoma tissue both at FL diagnosis and, as importantly, at transformation.

## 2. Results

The cohort included 26 males and 27 females with a median age at diagnosis of 54 years (range 35–78) (Table 1). Of these, 34 were nt-FL patients, while 20 were st-FL patients. Clinicopathological characteristics were similar in the two groups, although the st-FL patients presented with more adverse Ann Arbor stage and FLIPI scores. 

The immunohistochemical evaluation of IDO1 revealed cytoplasmic staining of cells mainly localized outside the follicular areas; see Figure 1A and Appendix A. All diagnostic FL biopsies exhibited some degree of IDO1 expression, although this varied substantially between the samples, with a median AF of 4.6% (range 0.007%–34.1%). There was no correlation with any of the analyzed clinicopathological features. 

According to the subsequent transformation status, no significant differences were seen in the IDO1 expression levels when comparing nt-FL samples (median 3.9%, range 0.007–21.9%) with st-FL samples (median 5.7%, range 0.01–34.1%; *p* = 0.185); see Figure 1B. The expression of IDO1 was also evident in all high-grade tFL samples (median 2.4%, range 0.01–12.0%). However, compared with paired diagnostic st-FL samples, no significant statistical correlation was found (*p* = 0.190); see Figure 1B. Accordingly, no differences in outcomes were observed in FL patients based on high versus low IDO1 expression; see Figure 1C–E. We previously analyzed the expression of another immune checkpoint inhibitor in the same study cohort, namely programmed death 1 (PD-1), which revealed prognostic value from strong PD-1 expression levels [18]. In the present study, the levels of IDO1 and strong PD-1 expression revealed a significantly positive correlation coefficient of rho = 0.3 (*p* = 0.031), thus indicating an intermediate positive correlation between the two immune checkpoint inhibitors.

## 3. Discussion

Previously, only a few studies have investigated IDO1 and the Trp/kyn pathway in FL, and none have included analyses regarding the transformation of FL. Masaki et al. [18] demonstrated significantly elevated kyn and lowered Trp levels in pretreatment serum samples from FL patients compared with healthy controls. In that study, high kyn levels correlated with poorer outcomes. In addition, these workers showed the presence of IDO1 staining in tumor microenvironmental cells in six FL samples [18]. Whether this finding mirrors the direct effect from the tumor itself suppressing the host antitumor immune response or reflects a host response remains unknown. Thus, additional and larger in-depth mechanistic studies are warranted to fully elucidate the possible role of the Trp/kyn pathway in FL.

Overall, the finding of IDO1 expression in all diagnostic FL biopsies suggests IDO1 as a potential novel therapeutic target in IDO-positive FL cases. IDO1 expression has been reported in several cancers in which IDO1-positive cells in the tumor microenvironment may facilitate tumor immune escape through the upregulation of Tregs [12,13]. Therefore, treatment strategies against IDO1 would not only directly target tumor cells, but also result in the downregulation of immunosuppressive signals. This rationale has provided the grounds for clinical trials testing IDO inhibitors and vaccination against IDO-derived peptides [13]. In a phase-1 trial to evaluate the efficacy and safety of IDO vaccines in advanced non-small-cell lung cancer (NCT01219348), a favorable median overall survival was observed, with no grade-3 or -4 toxicities. Furthermore, all the treated patients had a significantly reduced Treg cell population after the sixth dose of the vaccine [12,13,17]. Andersen et al. found IDO expression in the tumor microenvironment of non-small-cell lung cancer; however, the authors found no correlation between expression intensity and clinical response to the vaccination [13]. Thus, the presence of IDO alone was sufficient to elicit clinical responses. Furthermore, a phase-2 trial has been initiated investigating nivolumab therapy with the PD-1-targeting antibodies in combination with the programmed death ligand 1 (PD-L1)/IDO vaccine (NCT03047928). Thus, boosting specific effectors that recognize the two immune regulatory pathways may modulate immune regulation in a synergistic manner. Whether the PD-1/PD-L1 and IDO pathways cooperate has not been determined. However, in the present study, a correlation was found between tumoral IDO1 and PD-1 expression, indicating that the PD-1/PD-L1 immune checkpoint axis is relevant in the context of IDO1 signaling. Therefore, anti-IDO treatment, possibly in combination with other immunotherapeutic agents, may pose as a novel future treatment in FL. FL remains largely incurable, and additional personalized therapies are needed if we ever seek to ultimately overcome the disease. The presence of IDO1 in all cases suggests a possible target in FL treatment. Besides targeting the tumor cells themselves, enhancing the patient’s own immune system response might lead to a better response against the disease. With this study, we aimed at characterizing IDO1 expression in FL tissue. The choice of immunohistochemical analyses and the simple approach were based on easy implementation in clinical routine diagnostics at pathology departments if IDO1 ever are to be implemented as part of future FL treatment. More advanced studies must focus on understanding the mechanistic role of IDO1 through future in-depth mechanistic explorations. Due to the smaller number of samples included in the present study, larger and independent studies must be conducted evaluating the results. Thus, the data presented in our study provide a rationale for future investigations in larger and independent cohorts of FL and, presumably, high-grade lymphomas as well.

## 4. Materials and Methods

IDO1 protein expression was evaluated in diagnostic formalin-fixed, paraffin-embedded (FFPE) lymphoma tissues from 53 patients diagnosed with FL grade 1–3A at Aarhus University Hospital between 1990 and 2015; this cohort has previously been described [19,20,21,22,23]. Of these patients, 33 had no evidence of histological transformation with a follow-up of at least 10 years (non-transforming FL, nt-FL), while 20 patients experienced transformation to FL grade 3B or DLBCL at least six months after the initial FL diagnosis (subsequently transforming-FL, st-FL). For the latter, paired high-grade samples from the time of transformation were also included in the analyses (transformed FL, tFL). All cases were reviewed and reclassified according to the World Health Organization 2017 classification criteria for lymphoid tumors [1]. Reference [24] is cited in the Appendix A. The expression levels of IDO1 were quantified using immunohistochemical digital image analysis with the output as the area fractions (AFs) of IDO1-positive staining in relation to the total area of lymphoid tissue. The cutoff values for high versus low IDO1 expression were based on a Youden’s index and ROC analysis. The methods are described in detail in the Appendix A.

## 5. Conclusions

In summary, we report IDO1 expression in all cases of FL and tFL. IDO1 status was not a prognostic biomarker in relation to clinicopathological features or survival, nor did it predict patients’ risk of subsequent histological transformation. Nonetheless, IDO1 levels correlated with tumoral PD-1 expression, and the presence of IDO1 at the time of diagnosis provides grounds for future investigations of anti-IDO1 therapy as a possible treatment target of FL patients.

## Figures and Tables

**Figure 1 ijms-24-07314-f001:**
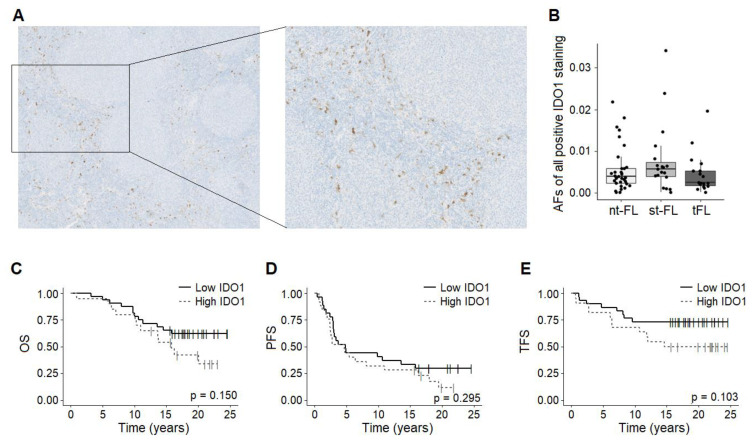
Intratumoral expression of IDO1 in FL. (**A**) Representative images of immunohistochemical staining of IDO1 in diagnostic nt-FL biopsies. Left: magnification of ×5, right: magnification of ×10. (**B**) Boxplot showing AFs of IDO1 in nt-FL, st-FL, and tFL samples, respectively. (**C**–**E**) Kaplan–Meier curves showing survival stratified according to IDO1 status and OS, PFS, and TFS, respectively. AF, area fraction; IDO1, indoleamine 2,3-dioxygenase 1; nt-FL, non-transforming FL; OS, overall survival; PFS, progression-free survival; st-FL, subsequently transforming FL; tFL, transformed FL; TFS, transformation-free survival.

**Table 1 ijms-24-07314-t001:** Patients’ clinicopathological features.

Characteristics	All n = 53 n (%)	nt-FL n = 33 n (%)	st-FL n = 20 n (%)	*p*-Value
Sex				NS
Male	26 (50)	15 (47)	11 (55)	
Female	27 (50)	18 (53)	9 (45)	
Age at FL diagnosis				NS
Median	54	54	54	
Range	35–78	35–76	40–78	
Ann Arbor stage				0.014
I-II	17 (26)	15 (44)	2 (10)	
III-IV	34 (71)	17 (53)	17 (85)	
Unknown	2 (3)	1 (3)	1 (5)	
FLIPI				0.012
Low	24 (37)	20 (59)	4 (20)	
Intermediate	18 (28)	9 (29)	9 (45)	
High	7 (29)	2 (6)	5 (25)	
Unknown	4 (6)	2 (6)	2 (10)	
LDH elevation				NS
Yes	2 (12)	1 (3)	1 (5)	
No	47 (82)	30 (91)	17 (85)	
Unknown	4 (6)	2 (6)	2 (10)	
B-symptoms				NS
Yes	12 (23)	6 (18)	6 (30)	
No	38 (72)	26 (79)	12 (60)	
Unknown	3 (6)	1 (3)	2 (10)	
Performance score				NS
<2	41 (75)	28 (82)	13 (65)	
≥2	9 (20)	4 (15)	5 (25)	
Unknown	3 (5)	1 (3)	2 (10)	
Bone marrow involvement				NS
Yes	14 (31)	6 (18)	8 (40)	
No	32 (55)	23 (71)	9 (45)	
Unknown	7 (14)	4 (12)	3 (15)	
Anemia				NS
Yes	4 (8)	1 (3)	3 (15)	
No	46 (88)	31 (94)	15 (75)	
Unknown	3 (5)	1 (3)	2 (10)	
FL histology				NS
FL grade 1–2	45 (86)	28 (85)	17 (85)	
FL grade 3A	8 (14)	5 (15)	3 (15)	

LDH, lactate dehydrogenase; FLIPI, follicular lymphoma international prognostic index; NS, not significant.

## Data Availability

Data will be available upon reasonable request.

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
