# Peer review of "IDO1 Protein Is Expressed in Diagnostic Biopsies from Both Follicular and Transformed Follicular Patients"

_ijms, 2023, doi:10.3390/ijms24087314_

Round 1

Reviewer 1 Report

It would be necessary to better describe the role of IDO and the implications of the target or the strong rationale for finding an anti-IDO therapy and the advantages it could bring.

The work is also descriptive only. It does not hypothesize a mechanism of action or the possibility or role (Immunomodulatory compared to CAR T? Bites?) that it could have in therapeutic combinations, in an already very broad context for follicular lymphomas.

It therefore lacks perspective and therefore the value of a description without perspective is reduced

Author Response

It would be necessary to better describe the role of IDO and the implications of the target or the strong rationale for finding an anti-IDO therapy and the advantages it could bring.

Response: Despite its broad expression in various tissues, only limited functional descriptions of IDO is available in the literature. As described in the introduction, IDO1 catalyzes the first step of tryptophan catabolism in the kynurenine pathway (ll. 46-48). This ultimately leads to immunosuppressive functions, e.g., though activation of regulatory T cells (Tregs), as also described (ll. 48-50). Not much is published on the role of IDO, and as such we have described the current published literature in our introduction. As anti-IDO therapies already are emerging, we have added a section in the introduction on this to clarify the possibility of a near future treatment option, also in follicular lymphoma.

“Thus, as anti-IDO therapies already are emerging, this might be a novel target in future treatment of FL” (ll. 62-63).

Based on the reviewer’s comment, we may not have elaborated enough on the advantages anti-IDO therapy could bring in the field of follicular lymphoma (FL). Although generally characterized by an indolent clinical course, FL remains an incurable disease. Despite advantages in FL treatment in the past centuries, there is a need for additional personalized therapies, if we ever want to cure this disease. Besides fighting the cancer cells with chemo/immunotherapies, enhancing the patient’s own immune system response may be the solution to ultimately overcome the disease (or at least keep it from progressing). As mentioned, targeting IDO may decrease inhibitory immune responses (such as Tregs). Hypothetically, this could improve the patient’s immune system response, which might lead to better response against the disease. We have elaborated further on this in the discussion and added the following section:

“FL remains largely incurable, and additional personalized therapies are needed, if we ever seek to ultimately overcome the disease. The presence of IDO1 in all cases suggests a possible target in FL treatment. Besides targeting the tumor cells themselves, enhancing the patient’s own immune system response might lead to a better response against the disease.” (ll. 194-199).

The work is also descriptive only. It does not hypothesize a mechanism of action or the possibility or role (Immunomodulatory compared to CAR T? Bites?) that it could have in therapeutic combinations, in an already very broad context for follicular lymphomas.

Response: Yes, the work is very descriptive. Initially, without prior knowledge of IDO1 expression in FL and tFL, we aimed to explore/characterize expression levels in the biopsies. Here, we also included transformed samples, to explore whether any differences were seen depending on transformation status, as this had not been investigated before in the literature. In this study, we found no difference depending on transformation status, nor any difference in outcome. However, the presence of IDO1 in all cases suggests unexplored therapeutic potential in this patient group. We do believe that we have hypothesized a mechanism of action for IDO-vaccination in the discussion section: “Overall, the finding of IDO1 expression in all diagnostic FL biopsies suggests IDO1 as a potential novel therapeutic target in IDO-positive FL cases. IDO1 expression has been reported in several cancers in which IDO1-positive cells in the tumor microenvironment may facilitate tumor immune escape through upregulation of Tregs[12,13]. Therefore, treatment strategies against IDO1 would not only directly target tumor cells, but also result in the downregulation of immunosuppressive signals.” (ll. 119-124). Furthermore, investigating the role IDO-treatment could have in therapeutic combinations would be a whole other study than what was the scope of this study. In the discussion section, we do mention other clinical studies that have analyzed IDO-therapy (e.g., in combination with other immunotherapy such as PD-1/PD-L1 checkpoint inhibitors). The present study provides the grounds for further investigation into IDO-therapy in FL – a patient group which definitely need additional personalized medicine in the hunt for a cure, as mentioned in the previous comment. We have further elaborated on this in the discussion section: “Therefore, anti-IDO treatment, possibly in combination with other immunotherapeutic agents, may pose as a novel future treatment in FL.” (ll. 193-194).

It therefore lacks perspective and therefore the value of a description without perspective is reduced

Response: Based on the above description of the aim and scope of our study, we do not think the study lack perspective but agree with the reviewer that this study does not provide the rationale for treating FL patients with anti-IDO therapy, merely do we show that IDO is present in both FL at diagnosis and as importantly at transformation. To clarify this, we have added the following in the introduction/discussion section: “Therefore, in the present study, we aimed to describe IDO1 expression in lymphoma tissue both at FL diagnosis and as importantly at transformation.” (ll. 63-65).

Reviewer 2 Report

Major:

1. There was no difference between all groups regarding IDO1 expression; no difference in outcome were observed in FL patients based on high versus low IDO1 expression. The results the authors have obtained here did not convince me the significance of IDO1 in FL.

Minor:

1. The title can be improved such that audiences can catch the main conclusions from the title.

2. The authors need to introduce all the types of FL mentioned somewhere in the manuscript, like the differences, as well as similarities and associated treatments.

3. In figure 1, representative images of IDO1 staining from all groups should be shown. Rectangle in Figure 1A should be revised.

4. The correlation between IDO1 and PD-1 needs to be extended in the manuscript.

Author Response

Major:

  1. There was no difference between all groups regarding IDO1 expression; no difference in outcome were observed in FL patients based on high versus low IDO1 expression. The results the authors have obtained here did not convince me the significance of IDO1 in FL.

Response: Initially, we aimed to explore/characterize expression levels in the biopsies. Here, we also included transformed samples, to explore whether any differences were seen depending on transformation status, as this had not been investigated before in the literature. In this study, we found no difference depending on transformation status, nor any difference in outcome. However, the presence of IDO1 in all cases suggests unexplored therapeutic potential in this patient group.

Explorative research will not always show significant differences between patient groups; however, we still believe that it is important to report these findings, especially as knowledge on IDO1 expression is lacking in the current literature. Furthermore, FL is a patient group that still lacks therapeutic options in the struggle to overcome the disease. The presence of IDO1 expression in all cases makes this axis a possible target for treatment in this patient group in which no cure exists. Thus, it does not seem that IDO1 plays a role in transformation of FL, however, it might play a role as a future treatment target in FL. This study is the first to describe IDO1 expression in FL and tFL cases, thereby providing the grounds for further research of IDO-therapy in FL, which has not previously been described. To clarify, that the main results of the study was to show IDO1 expression widely in both FL and tFL, we have rewritten the aim presented in the introduction: “Therefore, in the present study, we aimed to describe IDO1 expression in lymphoma tissue both at FL diagnosis and as importantly at transformation.” (ll. 63-65).

Minor:

  1. The title can be improved such that audiences can catch the main conclusions from the title.

Response: We agree. We have worked out a new title to better inform what the paper concludes.

IDO1 protein is expressed in diagnostic biopsies from both follicular and transformed follicular patients

  1. The authors need to introduce all the types of FL mentioned somewhere in the manuscript, like the differences, as well as similarities and associated treatments.

Response: Samples were included from patients diagnosed with FL grade 1-3A. Transformation were included as FL grade 3B or DLBCL. This was addressed in supplementary methods, however, we have now also included it in the main manuscript: “IDO1 protein expression was evaluated in diagnostic formalin-fixed, paraf-fin-embedded (FFPE) lymphoma tissues from 53 patients diagnosed with FL grade 1-3A at Aarhus University Hospital between 1990-2015; this cohort has previously been described[18-22]. Of these patients, 33 had no evidence of histological transformation with a follow-up of at least 10 years (non-transforming FL, nt-FL), while 20 patients experienced transformation to FL grade 3B or DLBCL at least six months after the initial FL diagnosis (subsequently-transforming-FL, st-FL).” (ll. 203-208).

Additionally, we have elaborated further on current treatment of both FL and tFL in the introduction, thereby highlighting the possible benefits of eventual future anti-IDO treatment: “Current treatment of FL and transformed FL generally include chemotherapy and/or immunotherapy, often in combination with CD20-targeting rituximab.” (ll. 60-62).

  1. In figure 1, representative images of IDO1 staining from all groups should be shown. Rectangle in Figure 1A should be revised.

Response: We have added a supplementary figure so that the manuscript provides representative examples of all three patient groups. Rectangles symbolize the area which is shown in higher magnification.

  1. The correlation between IDO1 and PD-1 needs to be extended in the manuscript.

Response: As mentioned in supplementary methods, spearman rank correlation were used to test for the correlation between IDO1 and PD-1 (ll. 361-362). This resulted in a significantly positive correlation (rho=0.3, p=0.031), which indicates an intermediate positive correlation between the two immune checkpoint inhibitors, as mentioned in the results section (ll. 81-84). This is already included in the manuscript.

Round 2

Reviewer 1 Report

I have read the author's answers, I don't think they fully satisfy the critical points of the article which remains descriptive and does not highlight the prospective value of the presence of IDO as a therapeutic target.

However, taking note of this, it is a further specification on follicular lymphoma which perhaps in the future could have a therapeutic destiny .... but which remains to be demonstrated.

Author Response

Response: As described in the first reponse, we completely agree that our results are descriptive regarding the presence of IDO1 in FL biopsies. To the best of our knowlege, this study is the first of its kind to describe IDO1 expression in FL and, surely, we agree that the role of IDO1 in FL pathogenesis and FL treatment should be further investigated. With this study, we aimed at characterizing IDO1 expression in  FL tumor tissue. The choice of immunohistochemical methodologies and the simple approach is based on an easily future implementation in clinical routine diagnostics at pathology departments if IDO1 ever are to be implemented as part of future FL treatment. More advanced studies must focus on understanding the mechanistic role of IDO1 in FL indeed, although out of the scope of the present study. However, future more in-depth mechanistic explorations would make up very interesting studies in the future. We have elaborated further on this in the discussion section, to make sure that these considerations are clearly stated (ll. 199-203).

Reviewer 2 Report

The authors addressed all my comments. 

Author Response

Thank you for a well review. Your comments were definitely worthful to our study. It is much appreaciated that you took your time reading our paper.